# Understanding the effectiveness of government interventions against the resurgence of COVID-19 in Europe

Mrinank Sharma [1,2,3,22✉], Sören Mindermann [4,22✉], Charlie Rogers-Smith[5], Gavin Leech [6], Benedict Snodin[3], Janvi Ahuja[3,7], Jonas B. Sandbrink[3,7], Joshua Teperowski Monrad [3,8,9], George Altman[10], Gurpreet Dhaliwal[11,12], Lukas Finnveden[3], Alexander John Norman[13], Sebastian B. Oehm [14,15], Julia Fabienne Sandkühler[16], Laurence Aitchison[6], Tomáš Gavenčiak[17], Thomas Mellan[18], Jan Kulveit[3], Leonid Chindelevitch[18], Seth Flaxman [19], Yarin Gal [4], Swapnil Mishra [18,20,23✉], Samir Bhatt [18,20,21,23✉] & Jan Markus Brauner [3,4,22,23✉]

European governments use non-pharmaceutical interventions (NPIs) to control resurging waves of COVID-19. However, they only have outdated estimates for how effective individual NPIs were in the first wave. We estimate the effectiveness of 17 NPIs in Europe's second wave from subnational case and death data by introducing a flexible hierarchical Bayesian transmission model and collecting the largest dataset of NPI implementation dates across Europe. Business closures, educational institution closures, and gathering bans reduced transmission, but reduced it less than they did in the first wave. This difference is likely due to organisational safety measures and individual protective behaviours—such as distancing— which made various areas of public life safer and thereby reduced the effect of closing them. Specifically, we find smaller effects for closing educational institutions, suggesting that stringent safety measures made schools safer compared to the first wave. Second-wave estimates outperform previous estimates at predicting transmission in Europe's third wave.

[1] Department of Statistics, University of Oxford, Oxford, UK. [2] Department of Engineering Science, University of Oxford, Oxford, UK. [3] Future of Humanity Institute, University of Oxford, Oxford, UK. [4] Oxford Applied and Theoretical Machine Learning (OATML) Group, Department of Computer Science, University of Oxford, Oxford, UK. [5] OATML Group (work done while at OATML as an external collaborator), Department of Computer Science, University of Oxford, Oxford, UK. [6] Department of Computer Science, University of Bristol, Bristol, UK. [7] Medical Sciences Division, University of Oxford, Oxford, UK. [8] Faculty of Public Health and Policy, London School of Hygiene and Tropical Medicine, London, UK. [9] Department of Health Policy, London School of Economics and Political Science, London, UK. [10] Manchester University NHS Foundation Trust, Manchester, UK. [11] The Francis Crick Institute, London, UK. [12] School of Life Sciences, University of Warwick, Coventry, UK. [13] Mathematical, Physical and Life Sciences (MPLS) Doctoral Training Centre, University of Oxford, Oxford, UK. [14] Medical Research Council Laboratory of Molecular Biology, Cambridge, UK. [15] University of Cambridge, Cambridge, UK. [16] University of Essen, Essen, Germany. [17] Independent scholar, Prague, Czech Republic. [18] Medical Research Council (MRC) Centre for Global Infectious Disease Analysis, School of Public Health, Imperial College London, London, UK. [19] Department of Mathematics, Imperial College London, London, UK. [20] Abdul Latif Jameel Institute for Disease and Emergency Analytics (J-IDEA), School of Public Health, Imperial College London, London, UK. [21] Section of Epidemiology, Department of Public Health, University of Copenhagen, Copenhagen, Denmark. [22] These authors contributed equally: Mrinank Sharma, Sören Mindermann, Jan Markus Brauner. [23] These authors jointly supervised this work: Swapnil Mishra, Samir Bhatt, Jan Markus Brauner. ✉email: mrinank.sharma@eng.ox.ac.uk; soren.mindermann@cs.ox.ac.uk; s.mishra@imperial.ac.uk; s.bhatt@imperial.ac.uk; jan.brauner@eng.ox.ac.uk

The first wave of the novel coronavirus, SARS-CoV-2, resulted in dramatic excess mortality across many European countries from approximately February to June 2020. Most of these countries implemented a suite of non-pharmaceutical interventions (NPIs), including business closures, school suspensions, and gathering bans[1–4]. Although partial control was achieved in the summer months, a second wave[5,6] of the epidemic followed the reopening of European societies, lasting approximately from August 2020 to January 2021. NPIs remain the primary tool for infection control in the short term[7], with vaccines set to reach only a minority of the global population in 2021[8] and vaccination delays in Europe. The need to identify the most effective interventions to control infections is further increased by waning population immunity[9] and new variants of concern (VOC) with higher transmissibility, severity or antigenic escape[10–12].

The effectiveness of NPIs in the first wave of COVID-19 has been studied extensively by relating the timing of NPIs to the epidemic's trajectory across different countries[1–3,13–16]. Fundamentally, the used statistical methods compare transmission in the presence and absence of NPIs. First-wave NPI effectiveness was measured relative to baseline contact patterns in the very early phases of the pandemic, where organisational safety measures and individual protective behaviours were lacking. For example, schools operated largely without safety measures before they were closed in the first wave; closing them thus reduced transmission considerably[2,13–15,17]. First-wave estimates can thus serve as proxies for how much transmission is associated with various areas of public and social life (if operated without safety measures and protective behaviours), and as a valuable starting point for NPI effects in the first wave of a potential future pandemic.

However, first wave estimates alone are likely inadequate to fully assess the impact of introducing or lifting NPIs during an ongoing pandemic. After the first wave ended, contact patterns did not return to the pre-pandemic normal (details in Supplementary Note 1.1). Individuals and organisations have adopted protective measures such as distancing, regular testing, and improved ventilation[18,19]. These changes likely made various areas of public life safer and thereby reduced the additional effect of strict bans or closures. For example, closing a school with various safety measures in place is expected to have a smaller effect on transmission than closing a pre-pandemic school. If organisational safety measures and personal protective behaviours stay in place, second-wave estimates are likely more similar to current, yet to be studied, NPI effects and thus more relevant to current policy decisions. Should safety measures and behaviours be loosened as the pandemic declines, NPI effect sizes are expected to change to levels between those seen in the first and second waves. In *Generalisation of NPI effectiveness estimates across time*, we further discuss this point and empirically assess how well first- and second-wave NPI effectiveness estimates generalised to the third wave.

In addition, governments require effectiveness estimates for the specific NPIs presently used. In the second and third waves, European governments implemented NPIs of finer granularity than identified in the first wave studies[1–3,13–16]. These include the closure of specific business sectors (gastronomy, retail and leisure venues), bans on gatherings of various small group sizes below 10, and nighttime curfews. Identifying their effects is crucial as they form the building blocks of both present infection control and reopening plans.

Here, we provide effect estimates for individual interventions during Europe's second wave of COVID-19. European countries typically implemented several NPIs concurrently, e.g. in grouped tiers[20–22]. Therefore, identification of individual NPI effects requires a multinational dataset, making use of the fact that different countries implemented different groups of NPIs at different times. We also require subnational intervention data as NPIs in the second wave were often implemented in specific regions or areas. National modelling would obscure local heterogeneity, not only in NPIs but also transmission timings[22,23] and socio-economic factors, leading to ecological fallacies[23] and biased effect estimates. A salient example is the infection heterogeneity preceding the second wave in the UK; the strong north/south divide would obscure localised increases in transmission when aggregated nationally.

Because existing NPI trackers lack granular subnational data and suitable fine-grained intervention definitions[24–27], modelling NPI effectiveness during the second wave requires a novel NPI dataset. We introduce a systematic categorisation of interventions across a randomised sample of 114 regions in 7 European countries (Austria, the Czech Republic, England, Germany, Italy, the Netherlands and Switzerland). We manually gather intervention data and ensure high data quality through several validation procedures.

To deal with the challenges of the second wave, we develop a semi-mechanistic hierarchical Bayesian model that is more widely applicable than previous models[1–3,13–16]. In particular, we account for unmodeled changes in transmission with a latent random walk and prevent artefacts from low case counts by allowing stochasticity. This enables the estimation of 17 individual NPI effects from case and death data. Since NPI effectiveness estimates can be sensitive to modelling decisions[16,28], we evaluate robustness to changes in the data, model, epidemiological assumptions, and potential unobserved confounding factors.

## Results and discussion

**The combined effect of all NPIs was smaller in the second wave than in the first.** Using a semi-mechanistic Bayesian transmission model with a latent stochastic process, we link NPI implementation dates to case and death data in each region and estimate intervention effect sizes expressed as percentage reductions in the (instantaneous) reproduction number $R_t$ (Fig. 2). The effect sizes in the second wave were considerably smaller than those estimated for the first half of 2020. All NPIs included in the study together reduced $R_t$ by 66% [95% CI: 61–69%], compared to median reductions of 77–82% in the first wave[1,2]. The difference between the waves is more pronounced if we consider the effectiveness of the most stringent set of NPIs actually implemented in each region, rather than the (hypothetical) combined effectiveness of all NPIs included in the study. The most stringent set of NPIs implemented in each region reduced $R_t$ by an average of 56% [95% CI: 40–64%], compared to 76–82% in the first wave, even though NPIs in the second wave were often similarly strict or stricter[1,2]. Finally, $R_t$ was reduced from an average maximum of 1.7 [95% CI: 1.4–2.4] to a minimum of 0.7 [95% CI: 0.5–0.8] across regions in the second wave, compared to an average maximum of 3.3–3.8 and a minimum of 0.7–0.8 in the first wave[1,2].

We believe that these differences between the waves can likely be explained by differences in pre-intervention contact patterns, safety measures, and personal protective behaviours (see "Introduction"). These changes likely made various areas of public and social life safer and thereby reduced the effect of strict bans or closures. The results underscore the importance of viewing NPI effectiveness relative to the counterfactual safety measures and behaviour in the absence of the given NPI. Several other factors seem less important but may have contributed to the difference in NPI effects. First, a build-up of population immunity likely does not explain the reduction in NPI

effectiveness: attack rates were low in our period of analysis[29] and the in- or exclusion of population immunity did not change the estimated effect sizes. Second, reduced adherence to NPIs in the second wave[30] may have played a role, although adherence seems much more relevant to restrictions for individuals (nighttime curfews, mask mandates and bans on private household mixing) than for organisations (closures of business sectors and educational institutions). Finally, in many countries, the ascertainment rate of cases was increasing during the first wave[31,32]. However, this is expected to *decrease* the effects estimated from the first wave, the opposite of what we find.

**A detailed assessment of interventions in Europe's second wave**. A key challenge for identifying the effects of individual interventions is that governments often implemented several NPIs simultaneously (Fig. 1). During the first wave, interventions were implemented within a short time window; for a given intervention and region, on average 83% of the other interventions in that region started in the same 10-day period[2]. In the second wave, NPI implementation was spaced out (Fig. 1), with only 23% of interventions starting in the same 10 day period. With enough data from the first wave, it was still possible to identify the effects of broad interventions; in the second wave, we can identify a more fine-grained set due to the increased temporal spacing combined with a larger and subnational dataset (9.2× more NPI implementations than the largest study that focused on Europe[2]). For each pair of NPIs that we are able to disentangle, on average we observed one without the other for 6969 days across all regions (with a minimum of 635 days for limiting household mixing in private to ≤10 attendees and to ≤30 attendees). However, we only show the combined effect of indoor and outdoor gathering bans (of various stringencies) since these comprise all six NPI pairs that score lowest on the aforementioned metric. Our estimates are robust to changes in data and model parameters (see below under "Robustness of estimates"), in contrast to studies on smaller datasets from the first wave[1,28], indicating[33,34] that the data are sufficient to overcome collinearity.

We find that business closures were particularly effective, with a combined effect of reducing $R_t$ by 35% [95% CI: 29–41%] (Fig. 2A). Closures of gastronomy (restaurants, pubs, and cafes) had a large effect on transmission with an estimated reduction in $R_t$ of 12% [95% CI: 8–17%], broadly in line with the increases estimated to have occurred as a consequence of the UK's "eat out to help out" scheme in August[35]. We find a similar effect for closing night clubs [12%, 95% CI: 8–17%], which were predominantly shut earlier than other businesses; this substantial effect size may reflect early second wave superspreading[36]. The combined effect of closing retail and close contact services (such as hairdressers and beauty salons) is also considerable [12%, 95% CI: 7–18%]. Assuming that much of the effect is due to retail, which is the more common type of business, this underscores the potential risks of brief but very numerous indoor contacts[37]. Closing leisure and entertainment venues such as zoos, museums, and theatres had a small effect [3%, −1 to 10%]. Closing businesses remains an effective measure to control infections; on the other hand, additional safety measures are likely needed to avoid significant transmission in retail and close-contact services, gastronomy and nightclubs, as they reopen.

As a broad intervention, we found that banning all gatherings, including 1-on-1 meetings, had a large effect: a 26% [95% CI: 18–32%] reduction in $R_t$. By recording the number of persons and households allowed to meet, we can understand the effectiveness of various thresholds. We found no evidence for

diminishing returns in the number of persons allowed to meet; in fact, the strictest thresholds had considerably larger effects than less strict ones. This result is consistent with previous studies on the English tier system—Tier 2, which limits gatherings to six people, had a small effect while Tier 3, which further limits gatherings to two people amongst other interventions, had a large effect[20,21,38]. The small effect associated with more lenient person limits (10 or higher) contrasts with estimates from the first wave, which commonly found bans on much larger gatherings to be effective[2,13,15]. The difference could be due to voluntary protective behaviours, which were absent pre-pandemic, such as avoiding crowds and distancing (Supplementary Note 1.1), but also due to limited adherence to rules on private mixing[39]. The results suggest that during an ongoing pandemic, infection control can no longer rely on reductions in transmission from banning gatherings with 10 or more people. Defining a "lockdown" policy as a ban on all gatherings and closure of all nonessential businesses, we estimate a total reduction in $R_t$ of 52% [95% CI: 47–56%].

Most countries adopted different limits for public gatherings and household mixing in private at various times. We can therefore begin to disaggregate the effect of these gathering types and examine their relative effects (Fig. 2B). We find that both gathering types contributed to reducing the transmission of COVID-19. While the total effect of banning all private mixing exceeds that of banning public gatherings, it seems that private mixing restriction was only effective at a strict threshold of two people allowed to meet. As discussed above, this could be due to a combination of low adherence, ongoing safety measures at gatherings, and individuals voluntarily avoiding crowds[18,19].

Observational studies of the first wave consistently found that closing all educational institutions was among the most effective NPIs[2,13–15,17]. In strong contrast, we find that this effect was small in the second wave [7%, 95% CI: 4–10%]. We conjecture that a combination of safety measures, behaviour changes, and epidemiological factors[40] in the education sector prevented large undetected clusters which may have developed in the first half of 2020[41–44]. Indeed, schools in Europe's second wave operated under safety measures that some other organisations lacked: symptom screening, asymptomatic testing, contact tracing, sanitising, ventilation, distancing, reducing group sizes, and preventing the mixing of groups[41,45]. Our results are consistent with agent-based and compartmental modelling studies which predict large decreases in transmission in schools upon implementation of multiple safety measures[46,47]. Further, the effects of closures on transmission outside of the educational institutions might provide an additional explanation for the observed differences between the waves. In the first wave, closures of educational institutions were among the first major NPIs implemented in most countries[1,2]. This may have signalled the gravity of the pandemic and prompted the general population to behave more cautiously, reducing subsequent case and death numbers. In the second wave, this signalling effect associated with school closures may have been smaller, as school closures due to COVID were usually not among the early NPIs (the early periods of closed schools shown in Fig. 1 are normal school holidays unrelated to COVID). In addition, there may have been changes in how school closures impact interactions outside of schools, such as parents of school children being forced to work from home.

We documented student presence separately for universities (or higher education) and schools (both primary and secondary) by recording their local term times, holidays, and closure dates in all 114 regions, as well as identifying regions without universities. However, the relative effects for closing only schools or only

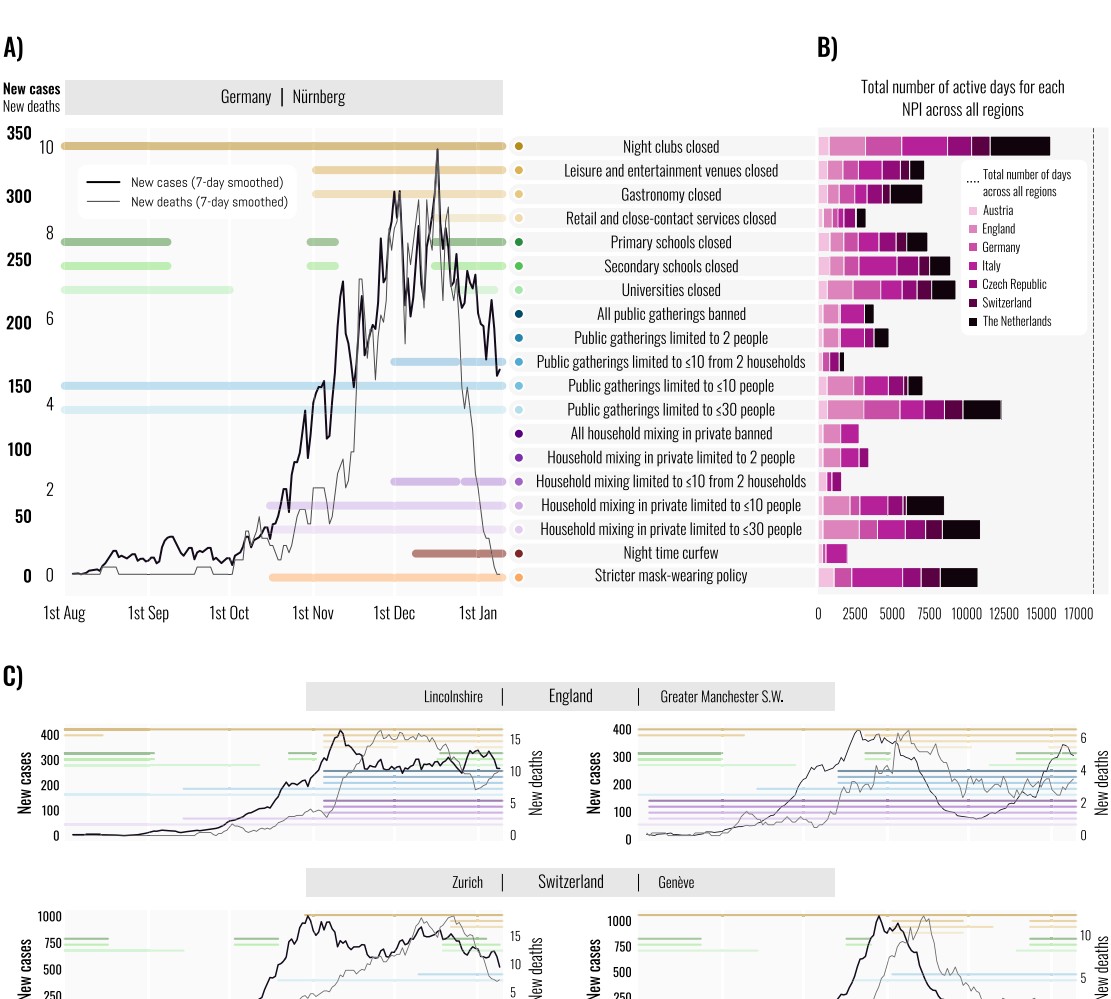

**Fig. 1 Dataset. A** Cases, deaths and implementation dates of nonpharmaceutical interventions in an example region (Nürnberg, Germany). Coloured lines indicate the dates that each intervention was active. Colours represent different interventions. **B** The total number of days that each intervention was used in our dataset, aggregated across $n = 114$ regions but separated by country. The dashed vertical line indicates the total number of region days in our dataset. **C** Additional timelines showing cases, deaths, and interventions in six regions. Comparing two regions within England (Lincolnshire and Greater Manchester S.W.) and within Switzerland (Zürich and Géneve) reveals significant subnational variation, both in the interventions used and in the evolution of the epidemic.

universities are not robust in a sensitivity analysis designed to adjust for undetected infections[48] in schools (Supplementary Fig. S10). We thus report the combined effect of closing all educational institutions, which is more robust.

Our findings underscore the impact of safety measures in educational institutions and support the view that school closures can be avoided if effective safety protocols are in place. However, safety measures vary by country[18] and further assessments are

required. Without sufficient measures, opening schools could lead to a resurgence[49]. In future pandemics, a promising strategy could be to close educational institutions early to gain time to implement safety measures, but then operate them throughout the pandemic whenever possible.

The introduction of policies that require mask-wearing in most or all shared/public spaces reduced transmission by 12% [95% CI: 7–17%]. Before the start of the second infection wave, countries in

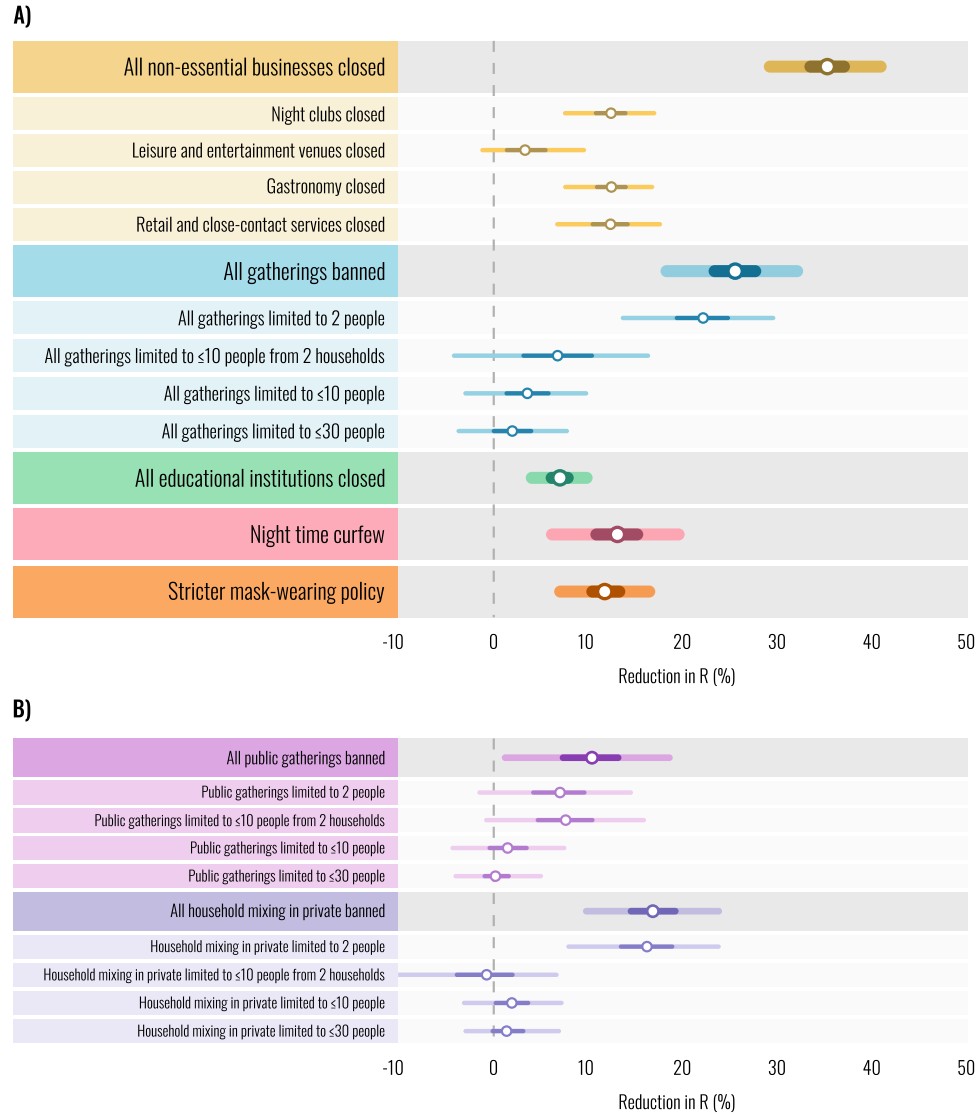

**Fig. 2 Intervention effectiveness under default model settings.** Posterior percentage reductions in $R_t$ shown. Markers indicate posterior median estimates from 5000 posterior samples across four chains. Lines indicate the 50 and 95% posterior credible intervals. A negative 1% reduction refers to a 1% increase in $R_t$. **A** Effectiveness of the main interventions included in our study. Intervention names preceded by "All" show the combined effect of multiple interventions. For example, "All gatherings banned" shows the combined effect of banning all public gatherings and all households mixing in private. **B** Individual effectiveness estimates for gathering types, separated into public gatherings and household mixing in private.

our dataset had less stringent policies that required mask-wearing only in select public spaces. The estimated effectiveness of this NPI is therefore the additional benefit of a stricter policy. In future epidemics with airborne pathogens, mandating mask-wearing in almost all, and not just some, public spaces early on will be an attractive strategy, given the comparatively low social and economic burden of this intervention.

Finally, nighttime curfews were commonly used in the second wave but have thus far received little study. In the countries in our dataset, they reduced transmission by 13% [95% CI: 6–20%], lending some evidence to their effectiveness as an infection control measure. Due to the broad nature of curfews and mask-mandates, these two interventions likely interact with other active NPIs and effectiveness may depend on the context. For example, a curfew may be less effective when all gatherings are already banned. In contrast, the other NPIs affect largely distinct areas of social activity and therefore are not expected to mutually interact to a great extent.

**Robustness of estimates**. The utility of our effectiveness estimates hinges on their robustness; estimates that are highly sensitive to modelling assumptions or confounding should not be used to guide policy. We perform 17 sensitivity analyses spanning 86 experimental conditions to evaluate robustness. Figure 3 shows how the median estimates of effect sizes from Fig. 2 vary across our sensitivity analyses, as we modify the priors and structure of the model, change the distributions of epidemiological delays, and randomly vary the set of regions and other aspects of the data. Since we cannot model all possible factors that affect the transmission, we also investigate sensitivity to unobserved factors[50] that influence $R_t$, acting as possible confounders. These include unrecorded NPIs and changes to ascertainment and fatality rates. Each analysis is shown in Supplementary Note 2.1. Supplementary Notes 2.2–2.7 describe additional validation experiments including multivariate sensitivity analysis, posterior predictive checks[51], simulation experiments, and a single-model meta-analysis across regions.

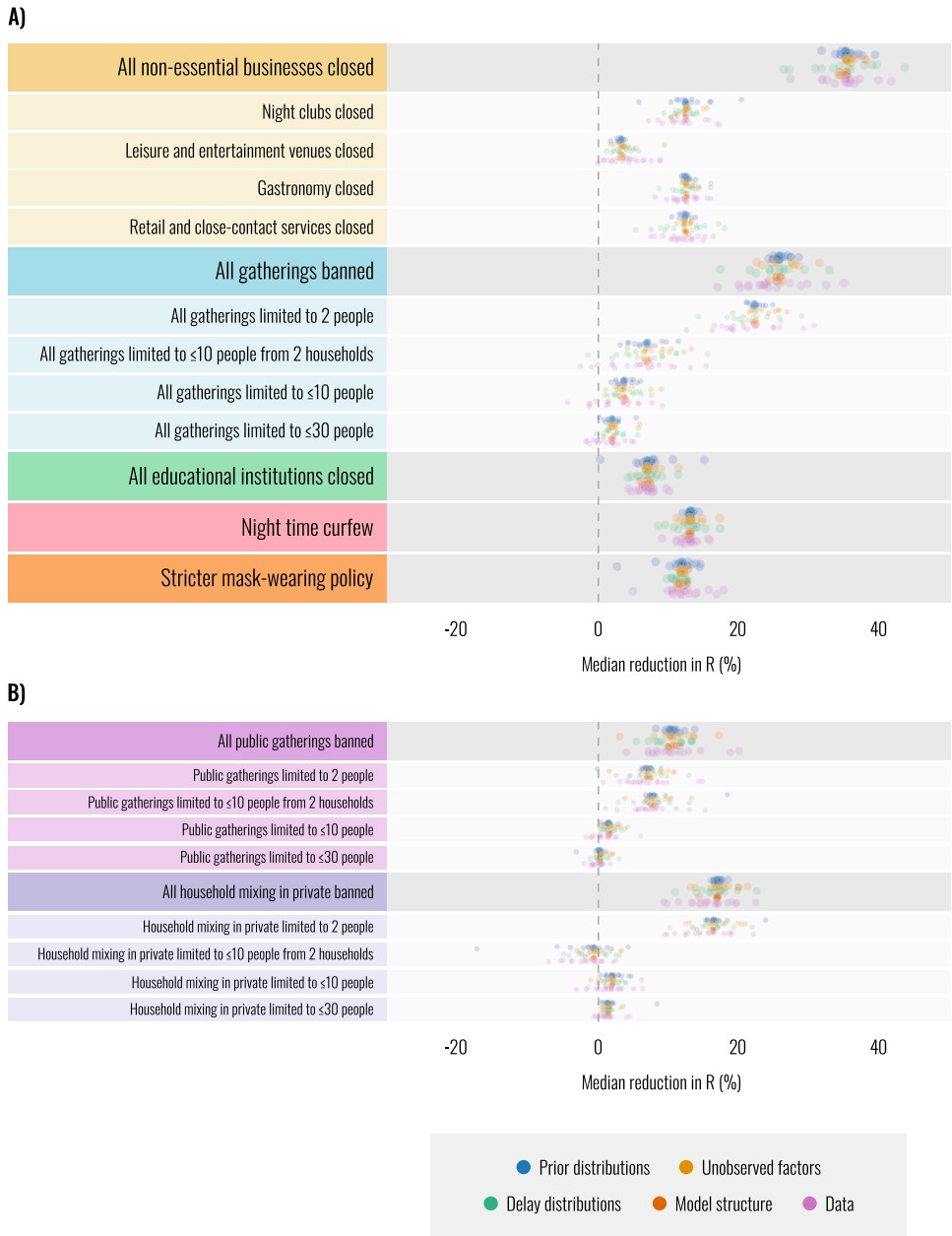

**Fig. 3 Robustness of median intervention effectiveness estimates across n = 86 experimental conditions (univariate sensitivity analysis).** Each dot represents the posterior median intervention effectiveness under a particular experimental condition. This figure contains only univariate sensitivity analysis—please see Supplementary Note 2.2 for multivariate sensitivity. Dot colour indicates categories of sensitivity analyses. Each category contains several sensitivity analyses (17 in total) and each sensitivity analysis contains several experimental conditions (n = 86 in total). Supplementary Table S1 lists all sensitivity analyses by category. **A** Robustness of effectiveness estimates of the main interventions included in our study. Intervention names preceded by "All" show the combined effect of multiple interventions. For example, "All gatherings banned" shows the combined effect of banning all public gatherings and all households mixing in private. **B** Robustness of the individual effectiveness estimates for separately banning public gatherings or household mixing in private.

While median NPI effects vary across the different experimental conditions, a broad picture emerges in which some NPIs outperform others across all experiments (Fig. 3). This suggests that high-level policy conclusions can be drawn from the results, as they depend on modelling assumptions only to a limited degree. Although our results are robust to varying strengths and types of unobserved factors, the true strength of unobserved confounding is unknown and our study is therefore subject to the limitations of observational approaches.

**A generalisation of NPI effectiveness estimates across time.** Empirical studies are limited to analysing past data, but NPI effects change in time according to a range of factors. Present policy decisions should therefore be informed by analysing periods of time as similar as possible to the current situation.

Since organisational safety measures and personal protective behaviours may account for much of the observed changes in NPI effectiveness, we expect past analyses to predict current NPI effects if similar safety measures and behavioural patterns are

observed today as in the analysed period. Indeed, safety measures and behaviours have been relatively stable throughout the second and third wave, and, at the time of writing, are far more widely adopted than in early March 2020 (details in Supplementary Note 1.1). For example, businesses incorporated measures to ensure minimum distances between persons; a measure not yet relaxed. Further, about two-thirds of YouGov survey respondents have consistently indicated that they avoided crowded places in the period from August 2020 to May 2021 (month-on-month average changes of less than 2%), compared to only 14% on 1 March 2020 (early data only available for the UK). Even in the UK, with low incidence and over 70% of the adult population vaccinated, safety measures and personal protective behaviours are considerably more prevalent today than in early March 2020 (Supplementary Note 1.1). At the time of writing, second wave estimates may therefore be more representative of current NPI effects than first wave estimates. In the coming months, governments can measure safety protocols and protective behaviours to inform which effectiveness estimates are appropriate. Should behaviours approach pre-pandemic levels as the pandemic declines, NPI effect sizes are expected to increase to levels between those of the first and second wave.

Novel VOCs[52] and increased population immunity due to vaccinations may affect overall NPI effectiveness. In addition, if new VOCs resulted in a higher initial $R_t$, then stricter or more NPIs would be required to bring $R_t$ below 1. If new VOCs were preferentially transmitted through certain demographics or activities, interventions targeting these would increase in effectiveness. Similarly, because vaccination campaigns prioritise older age groups, NPIs that primarily affect the young (school and university closures) can potentially achieve a higher relative reduction in transmission. Although vaccinated people might relax their protective behaviours, such behavioural changes only affect NPI effectiveness insofar as they occur in people who remain susceptible.

Finally, while we cannot experimentally test how well NPI effect estimates will generalise to future situations, we can assess how well estimates from the first and second wave have generalised to the third pandemic wave until now (Supplementary Note 1.2). In brief, we collected national NPI data for 6 European countries between January and May 2021, a period heavily influenced by the more transmissible VOC B.1.1.7 and increasing vaccination coverage. We then compared the observed changes in $R_t$ upon implementing/lifting NPIs to the changes predicted from first- or second-wave effectiveness estimates. The first-wave estimates were taken from previously published work[2]. The changes in $R_t$ predicted by first-wave estimates are on average 18 percentage points larger than the observed changes in $R_t$. In contrast, our second-wave estimates only overestimate the observed changes by 2 percentage points. Although these results are consistent with the aforementioned trends in safety measures and behaviours, this experiment has limitations. We measure the change in $R_t$ when NPIs are implemented/lifted, which may be affected by unobserved factors. For example, if an NPI is implemented at the same time as B.1.1.7 enters a country or an unrecorded NPI is lifted, we may observe an increase in transmission even if the NPI actually reduced transmission. As such, errors in prediction may not be due to errors in NPI effects.

**Implications**. European governments are presently debating which interventions to keep and which to remove. These are complex decisions that require weighing the clear social and economic costs of stringent measures against the damage from a continuously resurging and evolving epidemic. Our estimates provide a starting point to control infections in case of a

resurgence, but also to preempt virus evolution, which has spawned new variants in several areas where it appeared that the pandemic was overcome[10–12].

Using a European NPI dataset of unique scale and granularity with a flexible transmission model, we provide effectiveness estimates for individual NPIs in Europe's second wave. At a time when estimates from the first wave commonly form a basis for reopening plans[53], analysing NPI effects in the second wave reveals new conclusions to inform policy. We find that closures and bans still considerably reduced transmission in the second wave, but to a lesser degree than they did in the first wave. Estimates from the first wave overestimate NPI effectiveness in an ongoing pandemic because they measure the reduction in transmission compared to the pre-pandemic state where protective behaviours and safety measures were absent. Safety measures and behaviours likely made various areas of public and social life safer. If they stay in place, policymakers should not expect NPIs effects to be as large as they were in the first wave and should additionally refer to second-wave estimates to inform policy decisions. This is corroborated by experiments demonstrating that our NPI effectiveness estimates are largely unbiased estimates of the changes in $R_t$ that were observed in the third wave. Our results suggest that educational institutions, with appropriate safety measures, can be made considerably safer than they were before or early in the first wave; and that only the strictest limits on gathering size remain effective tools for infection control in an ongoing pandemic. In contrast, there is still considerable transmission associated with face-to-face businesses, and stricter mask-wearing policies and nighttime curfews can help curb transmission.

We note that we chose to express results as a percentage reduction rather than an additive reduction to ensure a property of diminishing returns to NPIs when the transmission is already low. This multiplicative model also naturally ensures positive reproduction numbers. Our results are based on a limited set of countries in the second wave. Expert judgement is thus needed to adjust them to local and contemporary circumstances.

The observation that NPI effectiveness is *dynamic* in time is an important and under-discussed consideration for policy. Our framework, which draws strength from a diversity of geographical localities and intervention timings, provides a systematic approach for both modelling and data collection. It can be used in near real-time and only requires routine case/death detection and the systematic identification of the relevant NPIs. It, therefore, generalises current approaches to real-time modelling except that the object of interest is not simply to summarise current transmission but also the factors driving it. To inform critical policy decisions, real-time modelling of evolving NPI effects should be a priority.

## Methods
### Data
*Dataset overview*. We collected a custom NPI dataset for this modelling study, as existing datasets do not provide sufficient geographical resolution to model the second wave (Table 1). Further advantages of our dataset are NPI definitions tailored towards the second wave and high data quality through extensive validation. All data necessary for the replication of our results are publicly available on https://github.com/MrinankSharma/COVID19NPISecondWave/tree/main/data, or, in archived form, at[54].

To create this dataset, we collected chronological data on NPIs that were in place between 1 August 2020 and 9 January 2021 in administrative regions, districts, and local areas of 7 European countries. The resulting dataset contains over 5500 entries on various NPIs in 114 regions of analysis (Supplementary Table S5). Every entry includes the NPI start date and end date, quotes and comments, and one or more sources from websites of governments and universities, legal documents, and/or media reports. Daily case and death data were obtained from government websites (Supplementary Table S4).

We now describe how we selected the countries, regions of analysis within each country, and NPI definitions.

| Table 1 Main dataset characteristics. | |
| --- | --- |
| Countries | 7 |
| Regions of analysis | 114 |
| Period | 1 August 2020–9 January 2021* |
| Days across all regions | 19,000 |
| NPI entries in the dataset | >5500** |
| Data validation (manual) | Semi-independent double entry***; interviews with local epidemiologists; validation against external sources; cross-country consistency checks |

In total, we collated >5500 intervention entries through a systematic categorisation.
*We ended the period of analysis before 9 January 2021 for English regions depending on their prevalence of a new variant of concern (see "Methods"). **Each entry includes the NPI start date and end date, quotes and comments, and one or more sources from websites of governments and universities, legal documents and/or media reports. ***Data were entered twice by two different groups of researchers. In the second round of data entry, researchers had access to the sources, quotes and comments found in the first round, but not to the NPI data entered in the first round (see *Methods*).

We first identified 7 European countries for which public data on daily reported cases and deaths were available at the same geographical resolution at which the country implemented NPIs (Austria, the Czech Republic, England, Germany, Italy, the Netherlands and Switzerland).

To gather initial information about the transmission-reducing NPIs used in these countries, we conducted an exploratory data collection and interviewed local epidemiologists from the countries. Based on these data, we created NPI definitions that faithfully represent the interventions that were implemented in these countries. We focused on clear-cut, major interventions that were implemented in many countries and we only recorded mandatory restrictions, not recommendations. We also accounted for closures that are not due to NPIs, such as vacation and term times in schools and universities, as we surmised that these effectively function as NPIs.

The exploratory data also informed the appropriate level of geographical granularity for the NPI data collection. In each country, we set our regions of analysis to correspond to the highest possible level of administrative division for which NPI implementations were identical throughout each region. The chosen administrative divisions were (Supplementary Table S4):

- States in Austria,
- Administrative regions in the Czech Republic,
- Nomenclature of Territorial Units for Statistics (NUTS) 3 statistical regions in England,
- districts in Germany,
- Administrative regions in Italy (with the exception of the Trentino-Alto Adige region, which was split into the autonomous provinces Trentino and Alto-Adige),
- Safety regions in the Netherlands, and
- Cantons in Switzerland.

For Austria, the Czech Republic, Italy and the Netherlands, it was feasible to collect data from the whole country (9, 14, 21 and 25 regions of analysis). From each other country, we took a stratified random sample of 15 regions of analysis. The sample was stratified by the regions' number of COVID-deaths in the first wave, to ensure a sufficiently diverse sample and reduce the variance of our NPI effect estimator. In Germany, each of the 16 German states had different regulations for its districts. To reduce the work required for data collection, we sampled the 15 districts only from the four largest states (Northrhine-Westphalia, Bavaria, Baden-Württemberg and Lower Saxony). These four states make up 60% of the population. Since regions with relatively few cases provide less evidence about the underlying reproduction number (and thus NPI effects), we increase statistical precision by excluding regions with fewer than 2000 reported cases during the analysis period.

*Data collection*. To ensure high data quality, the NPI data were collected with semi-independent double entry and several validation steps. Each country was collected by two authors of this paper, who were provided with a detailed description of the NPIs. The researchers manually researched all dates by using internet searches and screening (local) government press releases, ordinances and legislation. There was no automatic component in the data gathering process.

In the first round of data entry, the researchers initially collected the timeline of national NPI implementations. The researchers then compared their national timeline to the Oxford COVID-19 Government Response Tracker dataset[27] and, if there were any conflicts, visited all primary sources to resolve them. The data for each region of analysis were then entered by one of the two researchers, drawing on the national timeline and additional research. Several countries operated a tier or traffic light system that governed NPI implementation in subnational administrative divisions. For these countries, the researchers did not blindly enter the NPIs prescribed by the tier or traffic light system but additionally consulted local government websites and media reports to investigate if the NPIs prescribed by the national system were, in fact, implemented in a region of analysis.

In the second round of data entry, every entry was independently entered again by another researcher. This researcher had access to the sources found in the first round as well as the associated quotes and comments, but not to the NPI data

entered in the first round. This semi-independent double entry is similar to the validation used for parts of CoronaNet[24].

Finally, data from the two rounds of entry were compared and all conflicts were resolved by discussion and by visiting primary sources. A researcher then manually compared the data from all countries to ensure the consistent application of NPI definitions across countries. We also validated the data against further external sources (e.g.,[55] for Italy or[38] for England), contacted local epidemiologists when in doubt, and implemented a range of automated plausibility checks.

Throughout the data collection process, the researchers discussed edge cases and judgement calls on a shared online workspace to ensure consistency across countries. As expected, the validation process removed various sources of error and inconsistency. Supplementary Note 5 contains detailed explanations on coding decisions and judgement calls. The following software was used in the data collection and validation process: Google Chrome (various version numbers), Google Sheets (various version numbers) and Python/Numpy/Pandas for validation (various version numbers).

The total time spent on manual data collection, not including the design of the process, was 950 h, with 185 h on the national timelines, 470 h on collecting the regions of analysis, and 290 hours on the validation steps.

*Data preprocessing*. To mitigate bias, we excluded all observations in a region of analysis after the date when the VOC B.1.1.7 first made up >10% of infections in that region. Specifically, we excluded cases 5 days after >10% of all infections were due to the VOC and deaths 11 days after this value was reached. We chose these values to ensure that on the last included day, more than 80% of the reported cases and deaths were generated before the VOC exceeded >10% of all infections, according to our delay distributions. This only affected regions in England, usually towards the end of November[11].

The last day recorded in the intervention data set is 9th January 2021. Therefore, we included cases up to 5 days after this date and deaths up to 11 days after this date, as they are predominantly generated by infections before the 9th of January (see above).

Furthermore, to prevent influence from infections generated before the start of the analysis period, we excluded cases in the first 8 days (until 8th August) and deaths in the first 25 days (until 25 August). These values were chosen such that 80% of the cases and deaths recorded on the first observed day were generated in our window of analysis (including seeded infections), according to our delay distributions.

Supplementary Note 4.1 explains how we created the NPI features used in the modelling from the raw data. The final NPIs used for modelling are described in Table 2.

**Model**. We construct a semi-mechanistic Bayesian hierarchical model, similar to that of Brauner et al.[2], but with adaptations tailored for the second wave. Namely, we allow for changes in transmission unrelated to NPIs, which allow the model to explain e.g., unrecorded interventions. Furthermore, we account for the variance inherent in low incidence settings. Our model implementation is available on GitHub (https://github.com/MrinankSharma/COVID19NPISecondWave) or, in archived form, at[54].

We proceed by describing the model in Fig. 4 from bottom to top.

**Reproduction number**. The epidemic's growth is described by the time-and-location-specific (instantaneous) reproduction number $R_{t,l}$. $R_{t,l}$ is the expected number of secondary infections that would arise from a primary infection at time $t$ in location $l$, provided conditions remain the same after time $t$. We allow $R_{t,l}$ to change over time, even if the interventions implemented in location $l$ do not change. In particular, the value of $R_{t,l}$ depends on three factors: (a) the reproduction number at the start of the period in the absence of NPIs, $\widetilde{R}_{0,l}$; (b) the active nonpharmaceutical interventions (and their effectiveness); and (c) a latent (weekly) random walk. The random walk term allows $R_{t,l}$ to change from one week to the

**Table 2 NPI definitions.**

| NPI | Definition |
| --- | --- |
| Primary schools closed | Most or all primary schools (ages 5/6 to 10/11) have moved all teaching online or have closed (including for school holidays). |
| Secondary schools closed | Most or all secondary schools (ages 10/11 to 17/18) have moved all teaching online or have closed (including for school holidays). |
| Universities closed | Most or all higher education institutions are on (summer) term-break, (Christmas) vacation, or have sent students away from the university town (e.g., by closing university accommodation). As a result, a large fraction of students will have left their term-time accommodation to live at their home addresses. We did not count online teaching as a university closure if students were still expected to be present in the university town because (i) this still allows (likely considerable) transmission from students mixing outside of teaching events, and (ii) universities usually moved various components of their schedule online throughout the analysis period in a gradual manner. Some of the regions of analysis did not contain universities. For these, we counted universities as closed throughout the period of analysis. |
| Night clubs closed | Most or all nightclubs, discos, and other late-night venues are closed. |
| Gastronomy closed | Most or all gastronomy establishments/venues (restaurants, pubs and cafes) are closed or limited to take-away. |
| Leisure and entertainment venues closed | A large fraction of leisure and entertainment venues are closed. Common examples include theatres, cinemas, concert halls, museums, gyms, dance studios, indoor skating rinks, bowling alleys, public baths, indoor play areas, escape games, casinos, billiard rooms, zoos and amusement parks. |
| Retail and close contact services closed | All nonessential retail shops are closed. Only those retail shops designated as essential may open; common examples are supermarkets, pharmacies, and gas stations. In addition, all nonessential services that require close contact between customers and service providers are closed. This includes beauticians, nail salons, massage parlours, and—in all countries but Italy— hairdressers, but not medical services. |
| Nighttime curfew | Individuals must stay indoors during evenings/nights. There are exemptions for limited reasons, such as emergencies or caregiving. Whenever regions in our dataset introduced nighttime curfews, they essentially always also implemented, or already had in place, several other NPIs listed in this table (night clubs and gastronomy closure). These are encoded as distinct NPIs in the data. In our results, we thus estimate the additional effect of a nighttime curfew on top of other active NPIs[16]. |
| Stricter mask-wearing policy | Mask-wearing is required in most or all shared/public spaces outside the home (inside and outside) where other people are present or where social distancing is not possible. Already before implementing this policy, all countries in our dataset had some less strict policies in place that required mask-wearing only in select public spaces (see Supplementary Note 4.1). The estimated effectiveness of this NPI thus shows the additional benefit of the stricter policy. |
| Public gatherings limited to ≤30, ≤10, 2 people or banned. | Gatherings in public spaces are limited to a certain number of people. The limits of 30 and 6 include all regulations with at least that level of strictness. For example, a ban on public gatherings of more than 15 people would be classified as "public gatherings limited to ≤30 people". |
| Household mixing in private is limited to ≤30, ≤10, 2 people or banned. | Gatherings of individuals in private spaces are limited to a certain number of people. See the row above for additional explanations. |

next. Precisely, $R_{t,l}$ follows:

$$R_{t,l} = \underbrace{\widetilde{R}_{0,l}}_{\substack{R \text{ at } t=0 \text{ if no} \\ \text{NPIs active}}} \overbrace{\left(\prod_{i=1}^{I} \exp(-\beta_i \, x_{i,t,l})\right)}^{\text{effect due to active NPIs}} \underbrace{\exp(z_{t,l})}_{\text{latent random walk}} \quad ,$$

where $x_{i,t,l} = 1$ means NPI $i$ is active in location $l$ on day $t$ ($x_{i,t,l} = 0$ otherwise), and $I$ is the number of NPIs. We now explain each of these terms in more detail.

We place a prior distribution over $\widetilde{R}_{0,l}$, the reproduction number (in the absence of NPIs) on August 1st, 2020. In fact, many locations had some recorded interventions active at $t = 0$. Therefore, we chose the mean of the prior on $\widetilde{R}_{0,l}$ carefully. We ensured the prior on $R_{0,l}$ matched published estimates of $R_t$ for the first week of August from refs. [56] and [57]. For clarity, $\widetilde{R}_{0,l}$ is the reproduction number that would have been observed in location $l$ at $t = 0$ had no NPIs been

active. The prior over $\widetilde{R}_{0,l}$ follows:

$$\widetilde{R}_{0,l} \sim \text{Truncated Normal}\left(1.35, 0.3^2\right),$$

where truncation prevents values of $\widetilde{R}_{0,l}$ less than 0.1.

We parameterise the effect of NPI $i$ with the effect parameter $\beta_i$. This parameter is independent of time and shared across all locations, i.e., the effectiveness of a particular NPI is assumed to be identical across regions (though the random walk described below can account for differences). We place an Asymmetric Laplace prior over the effect parameter $\beta_i$, with scale parameter 30, asymmetry parameter 0.5, and location parameter 0. This prior has mean 0.05 and standard deviation 0.07. The prior allows for (unbounded) positive and negative effects as we cannot exclude the possibility that an NPI increases transmission. However, our prior places 80% of its mass on positive effects, reflecting a belief that NPIs are more likely to reduce transmission than to increase it. Furthermore, this is a shrinkage prior—it places more than 80% of its mass on "small" effectiveness (less than 10% change in $R_{t,l}$).

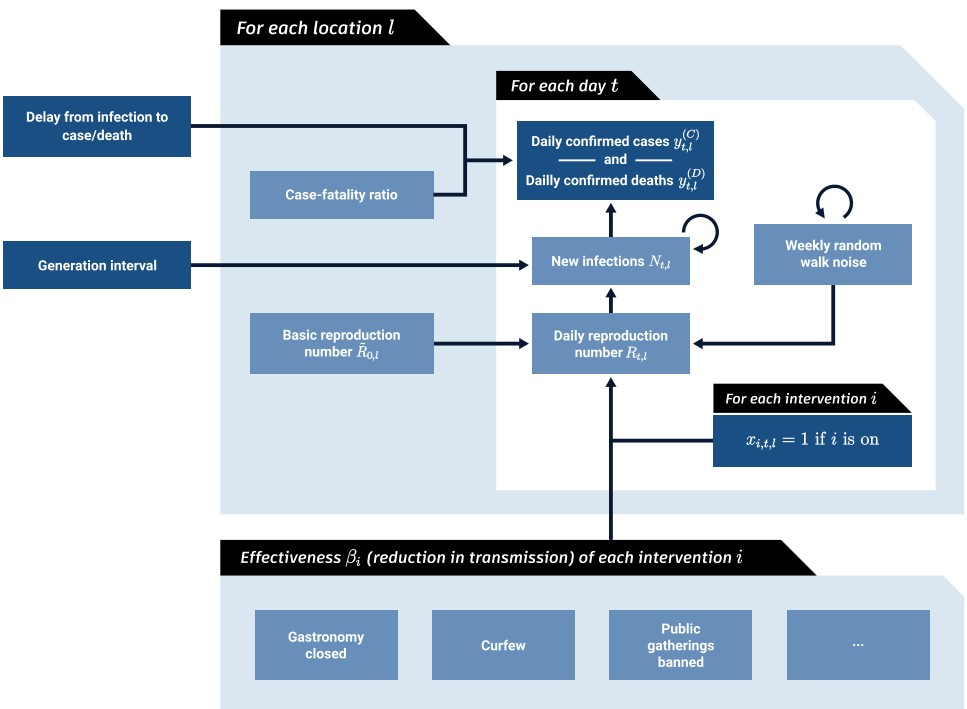

**Fig. 4 Model Overview.** Dark blue nodes are observed. We describe the diagram from bottom to top. The mean effect parameter of NPI $i$ is $\beta_i$. On each day $t$, a location's reproduction number $R_{t,l}$ depends on the basic reproduction number $\widetilde{R}_{0,l}$, the NPIs active in that location and a location-specific latent weekly random walk. The active NPIs are encoded by $x_{i,t,l}$, which is 1 if NPI $i$ is active in location $l$ at time $t$, and 0 otherwise. A random walk flexibly accounts for trends in transmission due to unobserved factors. $R_{t,l}$ is used to compute daily infections $N_{t,l}$ given the generation interval distribution and the infections on previous days. Finally, the expected number of daily confirmed cases $y_{t,l}^{(C)}$ and deaths $y_{t,l}^{(D)}$ are computed using discrete convolutions of $N_{t,l}$ with the relevant delay distributions.

The final component used to calculate $R_{t,l}$ is a location-specific latent random walk. This random walk allows for changes in $R_{t,l}$ every week that are due to factors outside the model. A random walk can explain lasting changes in transmission, unlike typical noise models. For example, suppose there was an unrecorded intervention in location $l$ at time $t$, or a recorded intervention with unusually low adherence. Then the random walk could be used to explain the observed change in transmission. Mathematically, the random walk noise terms follow:

$$z_{t,l} = \begin{cases} 0 & t \le 13 \\ z_{t-1,l} + \epsilon_{\lfloor (t-14)/7 \rfloor, l} & \text{if } t \bmod 7 = 0, \\ z_{t-1,l} & \text{otherwise} \end{cases}$$

where $\lfloor \cdot \rfloor$ denotes the floor operation and $\epsilon_{i,l} \sim \mathrm{Normal}(0, \sigma_R^2)$. In words, $z_{t,l}$ is set to 0 for the first two weeks, meaning that $R_{t,l}$ depends only on $\widetilde{R}_{0,l}$ and the active interventions for the first 2 weeks. Then, every week, the value of $z_{t,l}$ may increase or decrease depending on the noise variable $\epsilon_{i,l}$. If we observe that transmission increased in a particular week, then we may infer $\epsilon_{i,l} > 0$ and vice versa.

The random walk addresses an important limitation–we cannot include all possible factors that affect transmission. We can attempt to attribute effect sizes to NPIs at a time $t$, but we need to agnostically account for other unobserved factors that could have changed transmission (e.g., behaviour and adherence). By using a random walk, we include a latent stochastic process that agnostically models unobserved trends and residual structural correlations.

Furthermore, we place a prior over $\sigma_R$, which describes the scale of the random walk process. As $\sigma_R$ increases, the latent random walk can be used to explain larger changes in transmission. An advantage of placing a prior over $\sigma_R$ and performing joint Bayesian inference is that, if warranted by the data, an appropriate value may be inferred automatically. Our prior is $\sigma_R \sim \text{Half Normal}(0.15)$. We include this prior distribution in our sensitivity analysis (Supplementary Fig. S12) and find low sensitivity. Furthermore, we find that the data provide strong evidence about the value of $\sigma_R$ (see Supplementary Fig. S34 for a posterior and prior comparison).

**Infection process**. Let $N_{t,l}$ denote the number of new infections at time $t$ in location $l$. Furthermore, the generation interval (GI), which is the time between successive infections in a transmission chain, is denoted with the distribution $\pi_{GI}[\tau]$ where $\tau$ refers to the number of days since infection. The expected number

of infections then follows a discrete renewal process[58]:

$$\overline{N}_{t,l} = R_{t,l} \sum_{\tau=1}^{32} (\overline{N}_{t-\tau,l} \cdot \pi_{GI}[\tau]).$$

Renewal processes have a strong relationship to Hawkes processes and arise naturally from a Bellman Harris branching process[58,59]. The renewal equation has also been shown to be equivalent to a susceptible-exposed-infected-recovered Erlang model[60]. The renewal equation therefore specifies an epidemiologically motivated function class. One issue with the renewal equation is that it specifies a deterministic expectation for the number of new infections. This is generally suitable as infections become large, but in low incidence settings, estimation of $R_{t,l}$ can be sensitive to random fluctuations and noise. Therefore, we include an additive noise term, reflecting a belief that changes in the number of infections at low infection counts provide limited evidence to ascertain $R_{t,l}$, and must be treated with caution. Thus, the actual number of infections follows:

$$N_{t,l} = \text{softplus}(\overline{N}_{t,l} + \epsilon_{t,l}),$$

where $\epsilon_{t,l}^{(N)} \sim \mathrm{Normal}(0, \sigma_N^2 = 5^2)$. We use the softplus($\cdot$) rectifier to ensure that $N_{t,l} \ge 0$. See Supplementary Fig. S11 for sensitivity to the infection noise scale, $\sigma_N$.

We seed the model with one week of unobserved initial infections.

$$N_{-t,l} = \mathrm{Lognormal}(\widetilde{\mu} = 0, \widetilde{\sigma} = 3), \qquad \text{for } 1 \le t \le 7. \text{ Since we treat new}$$

infections as a continuous number, their initial value can be between 0 and 1.

**Infection ascertainment and fatality rates**. Scaling all values of a time series by a constant maintains its reproduction numbers. Our model is thus invariant to the scale of the observations and therefore to time-invariant differences between locations in the infection fatality rate (IFR), which is the proportion of infected people that subsequently die, and the infection ascertainment rate (IAR), which is the proportion of infected people who are subsequently tested positive. Since the model is invariant to the absolute scale of these rates, we set $\text{IAR}_l = 1$ for all local areas, and we place a prior over $\text{IFR}_l$. Both the IAR and IFR are assumed to be constant over time. In addition, since we assume $\text{IAR}_l = 1$, the IFR is actually a case-fatality rate and the variable $N_{t,l}$ effectively represents the infections that are later confirmed as positive cases. The uninformative prior over $\text{IFR}_l$ follows:

$$\text{IFR}_l \sim \text{Uniform}[10^{-3}, 1].$$

**Table 3 Epidemiological parameters, their distributional forms, and their sources.**

| Delay | Distributional form of delay | Source |
|---|---|---|
| Generation interval | Gamma(mean = 4.83, sd = 1.73) | Meta-analysis[67] |
| Incubation period | Gamma(mean = 5.53, sd = 4.73) | Meta-analysis[67] |
| Onset to reported death | Gamma(mean = 18.61, sd = 13.62) | Linelist |
| Onset to case confirmation | Gamma(mean = 5.28, sd = 3.75) | Linelist |

We then have:

$$N_{t,l}^{(C)} = N_{t,l}, \text{ and } N_{t,l}^{(D)} = \text{IFR}_l \cdot N_{t,l}.$$

As such, $N_{t,l}^{(C)}$ represents infections that are later confirmed, and $N_{t,l}^{(D)}$ represents infections that later result in death.

As part of our validation, we replace the assumed time-constant IFR and IAR with their estimates in England (applying these to all countries), taken from ref. [31]. These time-varying estimates of the IFR/IAR are estimated using seroprevalence data from ONS[61] and REACT[62], along with case and death time series for England. See Supplementary Fig. S24. We find that our NPI effectiveness estimates are not sensitive to this change.

**Observation model for cases**. The expected number of confirmed cases on day $t$ in location $l$ is given by a discrete convolution:

$$\bar{y}_{t,l}^{(C)} = \sum_{\tau=0}^{31} N_{t-\tau,l}^{(C)} P_C(\text{delay} = \tau),$$

where $P_C(\text{delay})$ is the distribution of the delay from infection to case-reporting. This distribution is truncated to 31 days for computational efficiency. As in prior works[1,2], the observed cases $y_{t,c}^{(C)}$ follow a negative binomial distribution with mean $\bar{y}_{t,c}^{(C)}$ and a country-specific inferred dispersion parameter, $\Psi_c^{(C)}$. Since different countries have different reporting practices, we allow $\Psi_c^{(C)}$ to differ by country. The prior over this parameter is as follows:

$$\Psi_c^{(C)} \sim \text{Half Normal}(5).$$

**Observation model for deaths**. The expected number of deaths on day $t$ in location $l$ is given by a discrete convolution:

$$\bar{y}_{t,l}^{(D)} = \sum_{\tau=0}^{63} N_{t-\tau,l}^{(D)} P_D(\text{delay} = \tau),$$

where $P_D(\text{delay})$ is the distribution of the delay from infection to death reporting. Similar to cases, the delay vector is truncated for computational reasons, but since the delay between infection and death is longer, we truncate this distribution to a maximum delay of 63 days.

Finally, the observed deaths $y_{t,c}^{(D)}$ follow a negative binomial distribution with mean $\bar{y}_{t,c}^{(D)}$ and a country-specific inferred dispersion parameter, $\Psi_c^{(D)}$:

$$\Psi_c^{(D)} \sim \text{Half Normal}(5).$$

Having separate dispersion parameters for cases and deaths ensures that they can be weighted differently if there is a difference in their output variance.

**Implementation**. The model was implemented in NumPyro (version 0.6.0)[63]. The model components in all previous equations are combined into a single likelihood function and a set of prior distributions. These ingredients are needed to infer a posterior over the unobserved variables in our model using the No-U-Turn Sampler (NUTS)[64], a standard Markov chain Monte Carlo sampling algorithm, as implemented in NumPyro. We used 4 chains with 250 warmup samples and 1250 draw samples, thereby obtaining 5000 posterior samples. We ensured that the posterior had converged by ensuring there were no divergence transitions, as well as monitoring the effective sample size and rank-normalised split-$\hat{R}$ statistic.

**Delay distributions—case and death delays**. Recall that our model requires external knowledge of the delay between infection and case confirmation as well as the delay between infection and death reporting. Many previous studies use estimates for delay distribution based on the data from the first wave[2,65]. However, these delay distributions may be different in the second wave due to sustained investment in testing capabilities and healthcare. Therefore, we re-estimate these delay distributions using data from the second wave.

The delay from infection to case confirmation is composed of the incubation period—the time from infection to onset of symptoms—and the symptom-to-confirmation delay. Similarly, the delay from infection to death reporting is composed of the incubation period and the symptom-to-death-reporting delay. We take an estimate of the incubation period from a meta-analysis[66]. We then combine this incubation period with estimates of the symptom-to-confirmation delay and

the symptom-to-death reporting delay from linelist data to form our total delay distributions.

We use linelist data from Austria, Germany and the United Kingdom (UK). This linelist data contains country-specific patient data of the date of symptom-onset, the date of case confirmation (for Austria, Germany and the UK) and the reported date of death (for Austria and the UK). To ensure that the linelist data we used was appropriate for the second wave, and to avoid censoring bias, we filtered the linelist data using the following conditions:

- Date of onset of symptoms $\geq 2020/07/01$
- Date of onset of symptoms $\leq 2020/11/01$
- Date of death $\leq 2021/01/22$
- Date of death $\geq$ date onset
- Date of admission $\geq$ date onset
- Date of test confirmation $\geq$ date onset

By neglecting symptom onsets dates past November, we mitigate censoring bias. There were almost 3 months since November for the latest possible onset date to fully evolve. Furthermore, by filtering the date of admission to be after the symptom-onset date, we prevent bias from hospital-acquired infections.

We fitted gamma distributions to the onset-to-confirmation and onset-to-reported-death data. We also fit Weibull, Lognormal, and Negative Binomial distributions to the data but, using model selection[67], found these to have an inferior fit. The fitted gamma distribution for the onset-to-confirmation delay has mean 5.28 days and standard deviation 3.75 days. The fitted gamma distribution for the onset-to-reported-death delay has mean of 18.61 days and standard deviation 13.62 days.

To compute the discretised delay vectors from infection to case confirmation, and for infection to reported death, we use Monte Carlo integration to discretise and sum the incubation period with the relevant delay.

**Delay distributions—GI**. We take an estimate for the GI from a meta-analysis[66]. We use Monte Carlo integration to discretise this delay.

Table 3 lists the delay distributions that we use, as well as their sources.

**Reporting summary**. Further information on research design is available in the Nature Research Reporting Summary linked to this article.

## Data availability

All data necessary for the replication of our results are publicly available on https://github.com/MrinankSharma/COVID19NPISecondWave/tree/main/data, or, in archived form, at[54]. The NPI data were collected by the authors; case and death data was taken from local data sources—please see https://github.com/MrinankSharma/COVID19NPISecondWave/blob/main/data/raw_data_w_sources/sources.md. National case and death data for the third wave experiments taken from John Hopkins University https://github.com/CSSEGISandData/COVID-19, but accessed the OxCGRT tracker https://github.com/OxCGRT/covid-policy-tracker.

## Code availability

All code necessary for the replication of our results, including reproducibility instructions, is available at https://github.com/MrinankSharma/COVID19NPISecondWave, or, in archived form, at[54]

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

## Acknowledgements

We thank Fabian Valka for advice on the Austrian COVID response, Ilaria Dorigatti for advice on Italy, Natalie Claire Ceperley for advice on Switzerland, Veronika Nyvltova for help with Czech NPI data, Paul Hunter for sharing his UK tier data, Toby Philips for advice on NPIs implemented in the second wave. We thank Muhammed Razzak for his comments on the paper.

M. Sharma was supported by the EPSRC Centre for Doctoral Training in Autonomous Intelligent Machines and Systems (EP/S024050/1) and a grant from the EA Funds programme. S. Mindermann's funding for graduate studies was from Oxford University and DeepMind. C. Rogers-Smith was supported by a grant from Open Philanthropy. A.J. Norman was supported by the U.K. BBSRC [grant number BB/T008784/1] and Open Philanthropy. J. Ahuja was supported by Open Philanthropy. J.T. Monrad was supported by the Augustinus Foundation, the Knud Højgaard Foundation, the William Demant Foundation, the Kai Lange and Gunhild Kai Lange Foundation, and the Aage and Johanne Louis-Hansen Foundation. G.Leech was supported by the UKRI Centre for Doctoral Training in Interactive Artificial Intelligence (EP/S022937/1). S.B. Oehm was supported by the Boehringer Ingelheim Fonds. L. Chindelevitch and S. Bhatt acknowledge funding from the MRC Centre for Global Infectious Disease Analysis (MR/R015600/1), jointly funded by the U.K. Medical Research Council (MRC) and the U.K. Foreign, Commonwealth and Development Office (FCDO), under the MRC/FCDO Concordat agreement; are part of the EDCTP2 programme supported by the European Union; and acknowledge funding by Community Jameel. S. Flaxman acknowledges the EPSRC (EP/V002910/1) and the Imperial College COVID-19 Research Fund. J.M. Brauner was supported by the EPSRC Centre for Doctoral Training in Autonomous Intelligent Machines and Systems (EP/S024050/1) and by Cancer Research UK. S. Bhatt acknowledges The UK Research and Innovation (MR/V038109/1), the Academy of Medical Sciences Springboard Award (SBF004/1080), The MRC (MR/R015600/1), The

BMGF (OPP1197730), Imperial College Healthcare NHS Trust—BRC Funding (RDA02), The Novo Nordisk Young Investigator Award (NNF20OC0059309), and The NIHR Health Protection Research Unit in Modelling Methodology. S. Bhatt thanks Microsoft AI for Health and Amazon AWS for computational credits.

## Author contributions

S. Mindermann, M.S., J.M.B., S. Mishra, S.B., Y.G., L.C., S.F. conceived the research. J.M.B., S. Mindermann, M.S., S.B., S. Mishra, B.S., G.A., J.A., L.F., J.B.S., G.D., J.T.M., A.J.N., J.K., G.L., J.F.S. designed and conducted the NPI data collection. M.S., S. Mindermann, S. Mishra, J.M.B., S.B., Y.G., G.L., S.F., L.A., L.C. designed the model and modelling experiments. M.S., S. Mishra, C.R.-S., G.L., and L.F. performed and analysed the modelling experiments. S. Mindermann, S. Mishra, J.M.B., S.B., J.T.M., J.K. did the literature review. S. Mindermann., S.B., J.M.B., M.S., S. Mishra, J.T.M., S.B.O., G.L., A.J.N., C. R.-S., B.S., G.D., G.A., J.A., J.B.S. wrote the paper. All authors read, gave input on, and approved the final paper. The listing order in the author list of M. Sharma and S. Mindermann was chosen at random.

## Competing interests

J. Kulveit has advised several governmental and nongovernmental entities about interventions against COVID-19. L. Chindelevitch has acted as a paid consultant to Pfizer and the Foundation for Innovative New Diagnostics, outside of the submitted work. He also volunteers as a scientist with the creative destruction lab Oxford. Y. Gal has received a research grant (studentship) from GlaxoSmithKline, outside of the submitted work. S. Bhatt sits on and advises the Scientific Pandemic Influenza Group on Modelling (SPI-M) a subgroup of the Scientific Advisory Group for Emergencies (SAGE). His work on this board is funded by the UKRI/MRC. The remaining authors declare no competing interests. None of the above-mentioned entities had any influence on the conceptualisation, design, data collection, analysis, decision to publish, or preparation of the paper.
