## [Peer Review File · Nature Communications]

Reviewers' Comments:

Reviewer #4:

Remarks to the Author:

The authors have performed additional analyses to address the questions raised by reviewers. Particularly, new analysis using recent survey data indicates that organizational safety measures and personal protective behaviors likely remained similar during the second wave. The authors also conditioned the generalization of the findings on the assumption that these behavioral patterns will last in the following waves. I think this is a better and more rigorous way to interpret the findings, though it limits the applicability of the findings in other settings. Technically, the analysis is sound and comprehensive.

I have one additional question regarding the generalization of the result. The effect of NPIs on disease transmission is formulated as a multiplicative impact on R_t (percentage reduction). The multiplicative model structure means the absolute R_t reduction depends on the initial R_t . For instance, if a measure is estimated to reduce R_t by 20% during the second wave, for which the initial R_t was 1.5, what is the expected effect during a future wave if R_t is initially 2.5 due to more transmissible variants? Would it be a 20% reduction or a 0.3 decrease in R_t ? How do we decide what's the combination of interventions needed to bring R_t below 1? Given control measures that are still in place, would the marginal effect further decrease due to diminishing return? An alternative is an additive model, but it also has its issues. A discussion on this point is warranted.

Reviewer #5:

Remarks to the Author:

Comments for authors, Nature Communications manuscript NCOMMS-21-23691-T "Understanding the effectiveness of government interventions in Europe's second wave of COVID-19".

General comments:

1. Many thanks to the authors for their careful revision. Most of the original concerns have been adequately addressed and only a few minor points remain. The addition of the new Section E on generalizability is particularly helpful in clarifying context for the results.

Specific comments (main text):

1. Page 2, line -1. "...that safer operation of schools is possible with stringent safety measures.". Does "safer operation" mean "safer than no restrictions"? It would be helpful to be a bit clearer on the comparative statement.
2. Page 7, last entry of right column of Table 1. How is the term "semi-independent" defined?
3. Page 17, line 4. The acronym "VOC" seems unfortunate since it very often refers to "volatile organic compound". If it is in common use for "variant of concern", no change required, but if the authors are coining the acronym, I suggest avoiding VOC.
4. Page 17, line 5. "If VOCs affect where transmission happens...". The term "where" seems to suggest some notion of geographic variation in risk, but this is not clear from the current wording. Does this statement refer to *geographic location of transmission* or to *geographic location of transmission of highly transmissible variants*? Would it be more accurate to say "If VOCs affect where transmission is more likely to occur..." (a statement regarding locally average rates of transmission across all circulating variants) or perhaps "If VOCs affect local risk of transmission..."?
5. Page 17, line -7 "They are thus largely unbiased estimates for overall changes in R_t upon implementing/lifting NPIs in the third wave". While I feel the main body of this paragraph largely

addresses the original concerns, this sentence is still a strong statement, and may be read as a statement of unbiasedness of the estimation procedure, rather than a check of potential biasedness of the specific estimates. May I suggest the following edit (or something similar): "As a result, we document close agreement between our second-wave estimates and observed due to implementing/lifting NPIs in the third-wave." I feel such an adjustment focuses on the small differences observed (an empirical assessment) rather than a statement that can be read as a conclusion on the unbiasedness of the estimation approach (a theoretical assessment). The further discussion in the conclusion makes this point clearer, but I recommend stressing the empirical nature of the confirmation here as well.

6. Page 18, line -9. "can be operated more safely". As in Specific Comment 1 above, I suggest specifying that this means "more safely than in prepandemic times".

Specific comments (methods):

1. Page 2, line -6. "To increase statistical power". Thank you to the authors for the updated information. I suggest replacing "statistical power" with "statistical precision". "Power" typically refers to a specific hypothesis testing setting ($\text{power} = 1 - \text{probability of a type II error}$) and our focus here is on estimation rather than testing. The authors' argument can pertain to both (increased statistical precision typically means increased statistical power for a hypothesis test), but the focus here is on estimation.

2. Page 9, line 9. "Erlang model". Do the references 11 and 12 cover the Erlang model? If not, I suggest adding a general reference here.

3. Page 9, line 10. "epidemilogically" to "epidemiologically".

4. Page 11. To finalize the Bayesian modeling, would it be possible to combine all of the model components to define the likelihood function (or to add a statement that the model components are combined to define the likelihood function which when convolved with the priors defined above provide the necessary ingredients to sample from the posterior distribution via NumPyro)?

Reviewer #6:

None

Point-to-point response

Response to Reviewer 4

The authors have performed additional analyses to address the questions raised by reviewers. Particularly, new analysis using recent survey data indicates that organizational safety measures and personal protective behaviors likely remained similar during the second wave. The authors also conditioned the generalization of the findings on the assumption that these behavioral patterns will last in the following waves. I think this is a better and more rigorous way to interpret the findings, though it limits the applicability of the findings in other settings. Technically, the analysis is sound and comprehensive.

I have one additional question regarding the generalization of the result. The effect of NPIs on disease transmission is formulated as a multiplicative impact on R_t (percentage reduction). The multiplicative model structure means the absolute R_t reduction depends on the initial R_t . For instance, if a measure is estimated to reduce R_t by 20% during the second wave, for which the initial R_t was 1.5, what is the expected effect during a future wave if R_t is initially 2.5 due to more transmissible variants? Would it be a 20% reduction or a 0.3 decrease in R_t ? How do we decide what's the combination of interventions needed to bring R_t below 1? Given control measures that are still in place, would the marginal effect further decrease due to diminishing return? An alternative is an additive model, but it also has its issues. A discussion on this point is warranted.

We thank the reviewer for taking the time to understand and critically evaluate our work and voice their concerns.

We think a multiplicative model is more mechanistically plausible from an epidemiological point of view. The main reason for this is what the reviewer has already noted – namely that intervention effect sizes should be relative to a starting point and contain a property of diminishing returns. To illustrate this, consider a stay-at-home order when R_t is close to zero, this should logically have little impact on R_t and this is achieved in a multiplicative model. However in an additive model, leaving aside the statistically problematic detail of negative R_t due to additive effects, the stay-at-home order will still have a large impact.

In the case of a variant with higher R_t and an NPI that (randomly) removes 20% of contacts, a multiplicative reduction of 20% is mechanistically expected because each infector has 20% fewer infectees available. To our knowledge most models in

infectious diseases are multiplicative, and this explains the choice of a Cox hazards model over Aalen's model.

Additionally, modelling NPI effects as a percentage change in R_t (or in the growth rate g), is standard practice, leading to more comparable results. It is present in the major mechanistic data-driven multi-NPI effectiveness studies (e.g. Hsiang et al., 2020, Banholzer et al., 2020.), as well as our previous work (Flaxman et al., 2020, Brauner et al., 2020)

Hsiang, S., Allen, D., Annan-Phan, S., et al., 2020. The effect of large-scale anti-contagion policies on the COVID-19 pandemic. *Nature*. <https://doi.org/10.1038/s41586-020-2404-8>

Banholzer, Nicolas, Eva van Weenen, et al., 2021. "Estimating the Effects of Non-Pharmaceutical Interventions on the Number of New Infections with COVID-19 during the First Epidemic Wave." *PloS One* 16 (6): e0252827. <https://doi.org/10.1371/journal.pone.0252827>

Flaxman, S., Mishra, S., Gandy, A., et al., 2020. Estimating the effects of non-pharmaceutical interventions on COVID-19 in Europe. *Nature*. <https://doi.org/10.1038/s41586-020-2405-7>

Brauner, Jan M., Sören Mindermann, Mrinank Sharma, et al. 2021. "Inferring the Effectiveness of Government Interventions against COVID-19." *Science* 371 (6531). <https://doi.org/10.1126/science.abd9338>

We have added this sentence to the section about VOCs:

Additionally, if VOCs resulted in a higher initial R_t , then stricter or more NPIs would be required to bring R_t below 1.

Note that this is true with a multiplicative model. E.g. if the initial R_t is 2, we only need to reduce it by 50%, but if the initial R_t is 4, we need to reduce it by 75%.

We have also added the following sentences to the discussion:

We note that we chose to express results as a percentage reduction rather than an additive reduction to ensure a property of diminishing returns to NPIs when transmission is already low. This multiplicative model also naturally ensures positive reproduction numbers.

Ultimately, to answer questions like the one raised by the reviewer precisely, we need to monitor NPI effects in real-time. Our method provides a flexible framework to do so, we we have noted in the last paragraph of the manuscript:

The observation that NPI effectiveness is dynamic in time is an important and under-discussed consideration for policy. Our framework, which draws strength from a diversity of geographical localities and intervention timings, provides a systematic approach for both modelling and data collection. It can be used in near real time and only requires routine case/death detection and the systematic identification of the relevant NPIs. It therefore generalizes current approaches to real-time modelling except that the object of interest is not simply to summarise current transmission but also the factors driving it. To inform critical policy decisions, real time modelling of evolving NPI effects should be a priority.

Response to Reviewer 5

General comments:

1. Many thanks to the authors for their careful revision. Most of the original concerns have been adequately addressed and only a few minor points remain. The addition of the new Section E on generalizability is particularly helpful in clarifying context for the results.

Thank you for the detailed and thoughtful comments. We did our best to fully address the previously raised points and agree that Section E was a useful addition to the manuscript. We address the new comments point-by-point below.

Specific comments (main text):

1. Page 2, line -1. "...that safer operation of schools is possible with stringent safety measures.". Does "safer operation" mean "safer than no restrictions"? It would be helpful to be a bit clearer on the comparative statement.

Thanks for alerting us to this ambiguity. As we explain in the Introduction, "*First-wave NPI effectiveness was measured relative to baseline contact patterns in the very early phases of the pandemic, where organisational safety measures and individual protective behaviours were lacking.*" Thus, "safer" here means "safer than in the very early phases of the pandemic", in which schools were indeed operated largely without restrictions.

We have changed the relevant sentence to:

Specifically, we find smaller effects for closing educational institutions, suggesting that stringent safety measures made schools safer compared to the first wave.

We had to choose this still short version here due to the strict word limit on the abstract, but we go into more detail in the main text (see e.g. below).

2. Page 7, last entry of right column of Table 1. How is the term “semi-independent” defined?

“Semi-independent” is defined in the Methods section:

“In the second round of data entry, every entry was independently entered again by another researcher. This researcher had access to the sources found in the first round as well as the associated quotes and comments, but not to the NPI data entered in the first round. This semi-independent double entry is similar to the validation used for parts of CoronaNet (24).”

The term “semi-independent” is used to contrast full independent double-entry, in which the data is entered independently by two (sets of) researchers with no information transfer between the two (sets of) researchers.

We added this explanation to the table:

****Data was entered twice by two different groups of researchers. In the second round of data entry, researchers had access to the sources, quotes, and comments found in the first round, but not to the NPI data entered in the first round (see Methods).*

3. Page 17, line 4. The acronym “VOC” seems unfortunate since it very often refers to “volatile organic compound”. If it is in common use for “variant of concern”, no change required, but if the authors are coining the acronym, I suggest avoiding VOC.

The acronym VOC is standard for “variant of concern”, see e.g. here or here. We thus kept the acronym.

4. Page 17, line 5. “If VOCs affect where transmission happens...”. The term “where” seems to suggest some notion of geographic variation in risk, but this is not clear from the current wording. Does this statement refer to *geographic location of transmission* or to *geographic location of transmission of highly transmissible variants*? Would it be more accurate to say “If VOCs affect where transmission is more likely to occur...” (a statement regarding locally average rates of transmission

across all circulating variants) or perhaps “If VOCs affect local risk of transmission...”?

Thanks. This was not intended to be a statement about geographic location. Instead, we meant something such as: If a new VOC is predominately transmitted among children, then closure of schools will become more effective.

We have changed the respective sentence in the manuscript as follows:

*Novel variants of concerns (VOCs) and increased population immunity due to vaccinations may affect overall NPI effectiveness. **If new VOCs were preferentially transmitted through certain demographics or activities, interventions targeting these would increase in effectiveness.***

5. Page 17, line -7 “They are thus largely unbiased estimates for overall changes in R_t upon implementing/lifting NPIs in the third wave”. While I feel the main body of this paragraph largely addresses the original concerns, this sentence is still a strong statement, and may be read as a statement of unbiasedness of the estimation procedure, rather than a check of potential biasedness of the specific estimates. May I suggest the following edit (or something similar): “As a result, we document close agreement between our second-wave estimates and observed due to implementing/lifting NPIs in the third-wave.” I feel such an adjustment focuses on the small differences observed (an empirical assessment) rather than a statement that can be read as a conclusion on the unbiasedness of the estimation approach (a theoretical assessment). The further discussion in the conclusion makes this point clearer, but I recommend stressing the empirical nature of the confirmation here as well.

In response to this comment, we have deleted the sentence in question.

6. Page 18, line -9. “can be operated more safely”. As in Specific Comment 1 above, I suggest specifying that this means “more safely than in prepandemic times”.

He have revised the sentence as suggested:

“Our results suggest that educational institutions, with appropriate safety measures, can be made considerably safer than they were before or early in the first wave;”

Specific comments (methods):

1. Page 2, line -6. “To increase statistical power”. Thank you to the authors for the updated information. I suggest replacing “statistical power” with “statistical precision”. “Power” typically refers to a specific hypothesis testing setting (power = 1 –

probability of a type II error) and our focus here is on estimation rather than testing. The authors' argument can pertain to both (increased statistical precision typically means increased statistical power for a hypothesis test), but the focus here is on estimation.

Good point. We have changed the wording accordingly.

2. Page 9, line 9. "Erlang model". Do the references 11 and 12 cover the Erlang model? If not, I suggest adding a general reference here.

We added the relevant citation:

Champredon, David, Jonathan Dushoff, and David JD Earn. "Equivalence of the Erlang-distributed SEIR epidemic model and the renewal equation." *SIAM Journal on Applied Mathematics* 78.6 (2018): 3258-3278.

3. Page 9, line 10. "epidemilogically" to "epidemiologically".

Thanks, fixed.

4. Page 11. To finalize the Bayesian modeling, would it be possible to combine all of the model components to define the likelihood function(or to add a statement that the model components are combined to define the likelihood function which when convolved with the priors defined above provide the necessary ingredients to sample from the posterior distribution via NumPyro)?
m

Yes, and indeed we did combine the model components to define a likelihood. To clarify this, we added the following statement:

The model components in all previous equations are combined into a single likelihood function and a set of prior distributions. These ingredients are needed to infer a posterior over the unobserved variables in our model using the No-U-Turn Sampler (NUTS) (65), a standard Markov chain Monte Carlo sampling algorithm.

Again, we would like to thank all reviewers for their time and effort.

Sincerely,

Mrinank Sharma, Sören Mindermann, Swapnil Mishra, Samir Bhatt, Jan Brauner, on behalf of the authors.